

# Exploring forearm muscle coordination and training applications of various grip positions during maximal isometric finger dead-hangs in rock climbers

Blai Ferrer-Uris[*], Denisse Arias, Priscila Torrado, Michel Marina and Albert Busquets[*]

Institut Nacional d'Educació Física de Catalunya (INEFC), Universitat de Barcelona (UB), Barcelona, Spain
[*] These authors contributed equally to this work.

## ABSTRACT

**Background.** Maximal isometric finger dead-hangs are used in rock climbing to strengthen finger flexors. Although various grip positions are often used when performing finger dead-hangs, little is known regarding how these grip positions can affect forearm muscle activity. Understanding how forearm muscles are recruited during dead-hangs could help foreseeing the potential for training of different grip positions. The aim of the present study was to explore the training applications of the various grip positions by comparing the activity of forearm muscles during maximal dead-hangs in rock climbers.

**Materials & Methods.** Twenty-five climbers performed maximal dead-hangs in three climbing-specific grip positions: CRIMP, SLOPE, and SLOPER. We recorded the maximal loads used and the sEMG of the flexor digitorum profundus (FDP), the flexor digitorum superficialis (FDS), the flexor carpi radialis (FCR), and the extensor digitorum communis (EDC). Individual and global (sum of all muscles) root mean square (RMS) and neuromuscular efficiency (NME) values were computed. Repeated measures analysis were performed to assess grip differences ($p < 0.05$).

**Results.** SLOPER showed the largest maximal load values among the three grip positions ($p < 0.001$, $d \geq 2.772$). Greater global ($p \leq 0.044$, $d \geq 0.268$), FDS ($p \leq 0.005$, $d \geq 0.277$), and FCR ($p < 0.001$, $d \geq 1.049$) activity was observed for the SLOPER compared to CRIMP and SLOPE, while EDC ($p \leq 0.005$, $d \geq 0.505$) showed lower activity in the SLOPER compared to the other two grip positions. SLOPER presented the highest global ($p < 0.001$, $d \geq 0.629$), FDP ($p < 0.001$, $d \geq 0.777$), FDS (only CRIMP *vs* SLOPER: $p < 0.001$, $d = 0.140$), and EDC NME ($p < 0.001$, $d \geq 1.194$). The CRIMP showed greater FDS activity ($p = 0.001$, $d = 0.386$) and lower NME ($p = 0.003$, $d = 0.125$) compared to SLOPE.

**Conclusions.** These results revealed that, under maximum intensity conditions, SLOPER could stimulate the FDS and FCR better than the other grip positions at the expense of using greater loads. Similarly, maximum CRIMP dead-hang could better stimulate the FDS than the SLOPE, even when using similar loads.

Corresponding author
Blai Ferrer-Uris, bferrer@gencat.cat

# INTRODUCTION

Rock climbing performance is highly dependent on the climber's ability to hang from minimal or hard-to-grasp surfaces or holds (*Bergua et al., 2018*). Furthermore, several studies suggested that the ability to apply force with the fingers (*i.e.,* finger strength) is critical in rock climbing performance (*Vigouroux, Goislard de Monsabert & Berton, 2015*; *Saul et al., 2019*; *Stien, Saeterbakken & Andersen, 2022*). When climbing, various types of grip position can be used depending on the surface or hold characteristics, such as its size, depth, and shape. However, the most frequently used grip positions among climbers are the crimp grip (CRIMP) and the slope grip (SLOPE) (*Schweizer, 2001*; *Quaine, Vigouroux & Martin, 2003*). The CRIMP (Fig. 1A) is executed with proximal interphalangeal (PIP) joints flexed at 90° and the distal interphalangeal (DIP) joints extended or hyperextended (*Schweizer, 2001*). In the SLOPE (Fig. 1B), the PIP is nearly extended and the DIP is flexed (*Schweizer, 2001*). The CRIMP is usually employed on small surfaces with sharp edges to increase the contact area between the fingertips and the hold, whereas the SLOPE is mostly used on larger holds (*Schöffl et al., 2009*). A particular version of the SLOPE is widely used when grasping a curved surface with increasing steepness, called a curved sloper (*Fuss et al., 2013*). When gripping a curved sloper (SLOPER), DIP and PIP joint positions are similar to those performed with the SLOPE, but the middle and the proximal phalanxes, and even the palm of the hand, are in contact with the hold (Fig. 1C).

Finger strength has usually been assessed in a reliable and valid way by measuring isometric finger actions while the climber hangs from holds with one or two arms, also known as finger dead-hangs (DH) (Fig. 1D) (*Bergua et al., 2018*; *Torr et al., 2020*). Similarly, DHs have been extensively used to specifically train finger strength (*López-Rivera & González-Badillo, 2012*; *López-Rivera & González-Badillo, 2016*; *López-Rivera & González-Badillo, 2019*; *Medernach, Kleinöder & Lötzerich, 2015*). When training finger strength, CRIMP or SLOPE grip positions are often used while manipulating the DH intensity by either varying the added load to the body mass or changing the deepness of the hold. It is important to account that high intensity actions of the finger flexors can put a lot of stress in the fingers and lead to acute or overuse injury events (*Vigouroux et al., 2006*; *Schöffl et al., 2009*; *Miro et al., 2021*). Finger injuries represent the highest injury prevalence in climbing (41% of all injuries), with annular pulley injuries being one of the main causes of injury in climbing (12% of all injuries) (*Lutter et al., 2020*). Annular pulleys are ligamentous structures that help preventing the finger flexors tendons from bowstringing away from the phalanxes (*Miro et al., 2021*). It seems that stress in these structures is increased when the interphalangeal joints are more flexed, especially the PIP, which causes an increase in the physiological bowstring of the flexor tendons (*Vigouroux et al., 2006*; *Schöffl et al., 2009*). In this sense, forces on the pulley system up to two to four times the applied force at the fingertip have been observed for the CRIMP grip, which can cause tensions in the pulley system close to the failure point (especially for the A4 pulley) (*Vigouroux et al., 2006*; *Schöffl et al., 2009*). Therefore, the SLOPE position could result in a safer way to train finger strength *via* DHs, especially in novice climbers. However, other parameters like the mode of muscle action (concentric *vs* eccentric) could also play a role

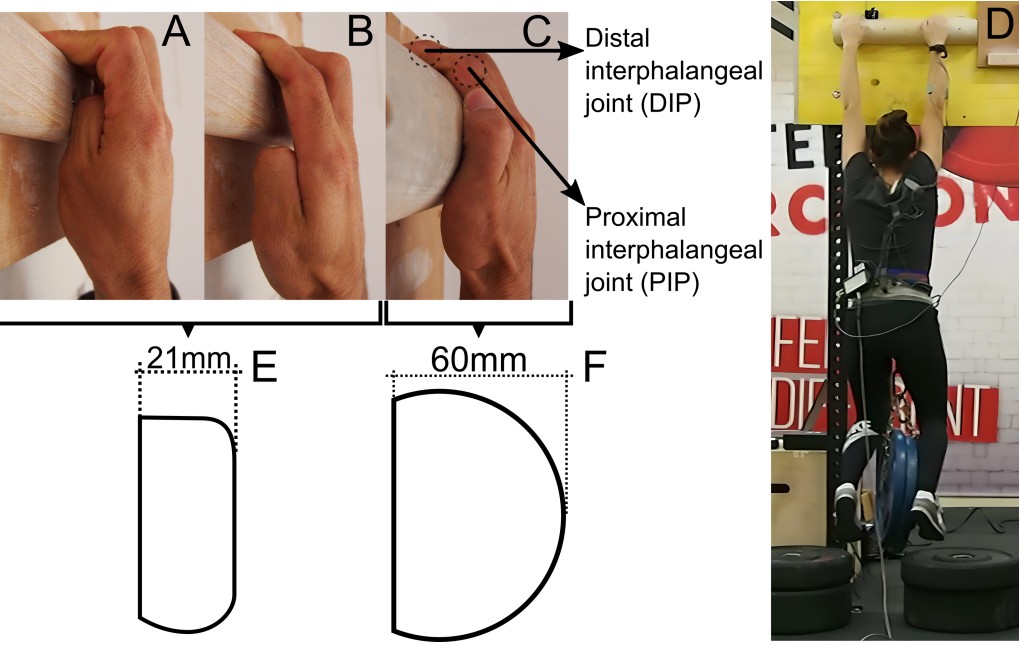

**Figure 1 Graphical representation of the grip positions and the climbing holds used during the finger dead-hangs.** The presented grip positions are: CRIMP (A), SLOPE (B) and SLOPER (C). The experimental setup during the dead-hang test with a participant hanging using the SLOPER grip is shown (D). Diagrams represent the climbing holds utilized for the CRIMP and SLOPE grips (E), and the SLOPER grip (F). In addition, distal and proximal interphalangeal joints are indicated in C.

in the risk of injury. It has been seen that risk of injury in the annular pulley system is higher when these structures are loaded while finger flexors act eccentrically, presenting injury events at lower fingertip forces, than when muscles act concentrically (*Schöffl et al., 2009*). During DHs, eccentric loading usually happens when the climber cannot hold the grip position because of fatigue, causing the hand to open (*i.e.,* fingers extend) while still applying great forces through the distal phalanx. This last phenomenon is possible because CRIMP and SLOPE are usually performed by contacting with only the distal phalanx with the hold, causing that the PIP and DIP joints need to be actively stabilized during the hang.

A possible way of minimizing these risks associated with finger injuries while training *via* DHs might be using a safe and stable grip position. SLOPER could result in low stress to the annular pulleys because it is naturally performed with the fingers in a rather extended position (like the SLOPE grip). In addition, SLOPER could present the added advantage that PIP and DIP joints might be passively stabilized during the hang because the three phalanxes are in contact with the hold, thus making it easier to maintain the grip form even when fatigued and preventing eccentric loading. Training finger strength using the SLOPER grip could be a great strategy to improve strength while minimizing risk of injury, a situation that could be especially convenient for novice climbers or during rehabilitation programs. Moreover, because the SLOPER implies a greater contact area with the hold, less skin tension would be expected. This could result in lower skin pain, abrasion, and splitting, which are not often seen as significant injuries but can have a great impact
in training volume and quality (*Shea, Shea & Meals, 1992*; *Bourne et al., 2011*). However, information about SLOPER grip is still scarce. Exploring the potential for training purposes while adopting these grip positions (CRIMP, SLOPE, and SLOPER), could be useful to guide finger strength training programs and weight the risks and benefits of using each one of them.

Understanding how forearm muscles are recruited during DHs could help foreseeing the potential for training of the abovementioned grip positions. Despite the interest in providing training measures and tests for strength assessment in climbing (*Bergua et al., 2018*), much less attention has been paid to forearm muscle activity during DHs. However, several climbing-related studies have assessed the forearm muscle activity *via* surface electromyography (sEMG) (*Quaine & Vigouroux, 2004*; *Watts et al., 2008*; *Dykes, Johnson & San Juan, 2019*). Most of these studies have focused on a sEMG analysis of wrist and finger extrinsic flexor and extensor muscles, which usually presented simultaneous activity or co-activity during gripping tasks (*Charissou et al., 2017*). Extrinsic flexor muscles of the fingers (flexor digitorum profundus, FDP, and superficialis, FDS) are usually reported as being the primary muscles involved during DHs and climbing (*Schweizer & Hudek, 2011*). Aside from finger flexors, other forearm muscles may also have an active role during climbing, like extrinsic wrist flexors (*e.g.*, flexor carpi radialis, FCR) and wrist and finger extensors (*e.g.*, extensor digitorum communis, EDC) (*Quaine, Vigouroux & Martin, 2003*; *Quaine & Vigouroux, 2004*; *Vigouroux, Goislard de Monsabert & Berton, 2015*).

Despite the paucity of information, it seems that using different grip positions could affect the participation of different forearm muscles during DHs and climbing. Tendon-force ratio in cadaver fingers and single-finger experiments presented evidences that CRIMP primary involved the FDP while FDS was mainly involved in the SLOPE (*Vigouroux et al., 2006*; *Schweizer & Hudek, 2011*). However, no study has assessed differences between these two grip positions under climbing-specific and intensity-equated conditions (*e.g.*, during maximal DH). Moreover, to our knowledge, no study has assessed differences in muscle activity between the SLOPER and the two abovementioned grip positions. Therefore, the aim of the present study was to explore the training applications of the CRIMP, SLOPE and SLOPER grip positions by comparing the muscle activity of the FDP, FDS, FCR, and EDC during maximal DHs.

## MATERIALS & METHODS

### Participants

Twenty-five participants (22 males and three females) (Table 1) were recruited for this observational cross-sectional study. Inclusion criteria were: (a) being 18 years of age or older; (b) having a minimum climbing experience of one year; (c) performing a minimum amount of regular climbing practice of one session/week in the previous 6 months; (d) self-report of a climbing grade equal to or higher than 6b on the French Rating Scale of Difficulty (*Draper et al., 2015*); (e) having a minimum experience of 4 weeks with DHs; and (f) being free of finger and upper body injuries. Rock climbing performance (*i.e.*, climbing grade) was converted to the International Rock Climbing Research Association (IRCRA) reporting

**Table 1  Participant characteristics.**

| Variable | mean ± SD |
|---|---|
| Age (years) | 29.0 ± 7.3 |
| Height (cm) | 170.7 ± 22.2 |
| Body mass (kg) | 75.1 ± 23.2 |
| Climbing experience (years) | 5.0 ± 3.3 |
| Training frequency (days/week) | 2.2 ± 1.1 |
| Climbing grade performance (IRCRA[a]) | 20.1 ± 3.2 |

**Notes.**
[a] International Rock Climbing Research Association reporting scale (*Draper et al., 2015*).

scale to compute descriptive statistics (*Draper et al., 2015*). This study was approved by the Ethics Committee for Clinical Research of the Catalan Sports Council (approval number: 25/2019/CEICGC) and was conducted in accordance with the Declaration of Helsinki. Written informed consent was obtained from all participants.

## Procedures
### Experimental setup
Two different 540 mm long climbing holds (Euro-Holds, Spain) were used for the DHs: (1) a rectangular hold with a 21 mm deep flat edge and a 10 mm rounded end (for the CRIMP and the SLOPE) (Fig. 1E) and (2) a half cylinder hold with a 60 mm deep radius (for the SLOPER) (Fig. 1F).

### Experimental protocol
All participants came twice to the laboratory. The first session served as familiarization, while the purpose of the second session was to test sEMG activity of the target forearm muscles when performing maximal DH using the three grip positions. The procedure for these two sessions was the same and it was designed to minimize the number of trials necessary to obtain maximal DH during the testing session. Both sessions started with participants performing a standardized warm-up. Next, participants performed the DH tests. DHs test protocol was repeated for all three grip positions (CRIMP, SLOPE, and SLOPER). Grip positions were presented in random order for the warm-up and the DH tests in both sessions.

   During the testing session, participants were asked to perform two valid repetitions with their maximum load of the DH test for each grip position. Participants' body mass (BM), height, loads, and sEMG activity in the DH tests were recorded during the testing session. The testing session was performed between 3 and 15 days after familiarization. Participants were asked to refrain from any training or climbing activity the day before.

### Warm-up protocol
Warm-up consisted of three sets of five repetitions of 5 s submaximal DHs (self-adjusted intensity) interspersed with resting periods of 15 s between repetitions and 1 min between sets. Similar DH warm-up protocols have been proposed previously (*Bergua et al., 2018*). Participants were instructed to "self-adjust the warm-up intensity for their fingers avoiding

forearm pump or fatigue". A different grip position (CRIMP, SLOPE, or SLOPER) was used randomly in each set.

### Dead-hang (DH) test protocol

The aim of the DH test was to determine the maximal load with which the participant could maintain a DH for 5 s, thus equating relative intensity across grips. Increasing loads were used by adding at least 1 kg between trials, according to the participant's self-perceived capacity. To avoid excessive fatigue, we tried to use less than five load increments per grip position until reaching the maximal load. The maximal load was considered as the last load increase that participants could hold for 5 s, maintaining the desired grip positions without losing finger contact with the hold and maintaining a straight arm position. Rests of three minutes were provided between trials and five-minute rests were given between grip positions to minimize fatigue effects across grip positions. The added load was attached to a climbing harness using a strap buckle and a carabiner. The holds were brushed between trials and sessions in order to maintain similar grip conditions across trials and participants, who were also provided with chalk to dry their hands before each trial. Participants were instructed to simultaneously hang with both hands, maintaining the instructed grip position without engaging the thumb in the grip: CRIMP, with 90° flexion at the PIP; SLOPE, 160–180° flexion at the PIP; or SLOPER, with full contact of the three phalanges with the sloper hold. In addition, participants were required to hang with elbows extended and engaged shoulders at a 180° flexion position, following recommendations from previous research (*Baláš et al., 2014*). Shoulders, elbows, and hands positions were carefully supervised by an experienced researcher. Only those attempts that were performed with engaged and 180 flexed shoulders, straight elbows, and maintaining the instructed grip position were considered valid. In addition, only for the testing session, to double check the grip position maintenance, a LifeCam HD-3000 webcam (Microsoft, Redmond, WA, USA) recorded the participants' hands during each attempt. Similar DH testing protocols have been used in previous studies showing that DHs are a good mean to measure climbing-specific finger strength in a valid and reliable way (*Bergua et al., 2018*; *Michailov et al., 2018*; *Torr et al., 2020*; *Stien, Saeterbakken & Andersen, 2022*).

## Data collection and variables
### Maximal isometric strength

During the DH test, the maximal added mass used by the participants for each of the grip positions was recorded and the maximal load (BM + the maximal added mass) and the relative maximal load (*i.e.,* maximal load/BM*100) were calculated for all the grip positions.

### Surface electromyography

The sEMG signals were recorded at 1000 Hz using a DataLog type no. P3X8 USB (Biometrics Ltd., Newport, UK) and SX230 sEMG sensors from the same manufacturer, which consisted of bipolar Ag-AgCl surface electrodes (10 mm diameter, 20 mm center-to-center distance) and a differential amplifier (gain 1000, input impedance 100 M $\Omega$, an input noise <5 $\mu$V, common mode rejection ratio higher than 96 dB). Participants' skin was shaved, abraded,
and cleaned with alcohol. Then, surface electrodes were secured with double-sided tape on the flexor digitorum profundus (FDP), flexor digitorum superficialis (FDS), flexor carpi radialis (FCR), and extensor digitorum communis (EDC) (Fig. 2) of the participant's self-reported dominant forearm. The electrodes on the FDS were placed approximately at 3/4 of the forearm length, slightly ulnarly on the line from the biceps tendon to the middle of the wrist. For the FDP, the electrodes were placed on the prominent bulge of the muscle at approximately 5–8 cm from the olecranon, slightly ulnarly and on the line between the olecranon and the lunate. For the FCR, electrodes were placed at around five cm from the medical epicondyle of humerus on the line between the medial humeral epicondyle and the proximal end of the second metacarpal. The EDC electrodes were placed around the 1/4 point on a line drawn from the lateral epicondyle to the styloid process of the ulna. The reference electrode was placed over the styloid process of the ulna. Electrodes were positioned on each muscle along a line between the origin and insertion of the muscle in the supinated hand, parallel to the direction of muscle fibers, determined using an anatomical atlas (*Perotto et al., 2011*), previous publications (*Matthews & Miles, 1988*; *Vigouroux & Quaine, 2006*; *Dykes, Johnson & San Juan, 2019*), and manual palpation. To check electrode placement and minimize crosstalk between electrodes, isolated contractions of each muscle were performed and electrode placement was adjusted when necessary. Flexion of the DIP of the fourth finger with immobilized PIP and neutral wrist position was performed to check the FDP electrode placement. Flexion of the PIP of the third finger with immobilized DIP and neutral wrist position was used to check the FDS electrode placement. Flexion of the wrist while maintaining extended fingers was used to check the FCR electrode location. Finally, extension of the fingers while maintaining a flexed wrist position was used to check the location of the EDC electrodes. Electrode placement was considered satisfactory when a clear and isolated activity of each of the muscles was obtained. In addition, when possible, inter-electrode distance between muscles was six cm or more, to minimize any possible crosstalk effect (*Mogk & Keir, 2003*).

sEMG data were analyzed using Spike 2.0 software (Cambridge Electronic Design Ltd., Cambridge, England). Raw sEMG data were filtered with a Butterworth Band-pass filter at 20–460 Hz. Root mean square (RMS) values were computed using a time window of 2 s of contraction, 1 s after the trial start. The average RMS for each muscle and grip position was computed between the two maximum DH trials ($RMS_{FDP}$, $RMS_{FDS}$, $RMS_{FCR}$, and $RMS_{EDC}$). Global muscle activity ($RMS_{Global}$) for each grip position was computed as the sum of the RMS of all muscles. In addition, neuromuscular efficiency (NME) was computed dividing the held force by the neuromuscular activity of each muscle (maximal load * 9.8/RMS) (*Magalhães et al., 2016*): $NME_{Global}$, $NME_{FDP}$, $NME_{FDS}$, $NME_{FCR}$, $NME_{EDC}$.

## Statistical analysis

Data normality was checked *via* exploration of histograms and by the Shapiro–Wilks test. Variable transformation was used when necessary. One-way ANOVAs with repeated measures or Friedman's test were used to test differences among grip positions. If a significant grip main effect was found in the ANOVAs or Friedman's test, pairwise

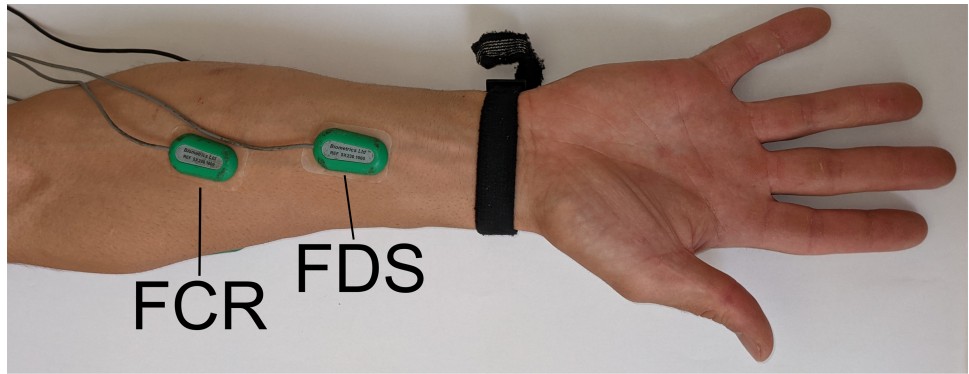

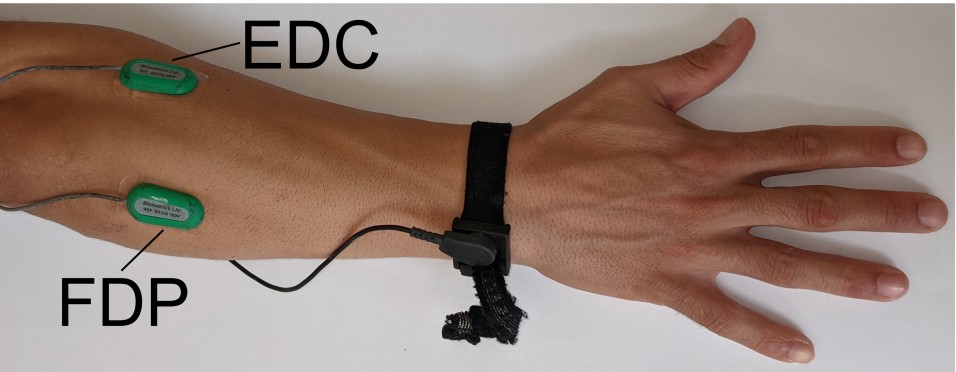

**Figure 2 Electrode location on the anterior and posterior face of the forearm.** FDP, flexor digitorum profundus; FDS, flexor digitorum superficialis; FCR, flexor carpi radialis; EDC, extensor digitorum communis.

comparisons were carried out between grip positions. Bonferroni correction was used for the ANOVAs and pairwise comparisons. Statistical analysis was performed using SPSS.21. The level of significance was set at $p = 0.05$.

The effect size of the different tests was calculated according to Cohen (*Cohen, 1988*): $d$ for $t$-test (0.2 small, 0.5 medium, and 0.8 large effect), $\eta_p^2$ for ANOVAs (0.01 small, 0.06 medium, and 0.14 large effect), and Kendall's W for Friedman's test (0.2 small, 0.5 medium, and 0.8 large effect).

## RESULTS

Descriptive analyses (mean and standard deviation) of muscle activity and Maximal and Relative maximal loads for each grip are presented in Table 2.

ANOVAs results (Table 3) revealed a significant grip effect among the three grips on Maximal load and Relative maximal load, showing that SLOPER allowed participants to hold greater Maximal loads (CRIMP *vs* SLOPER: $p < 0.001$, $d = 2.772$; SLOPE *vs* SLOPER: $p < 0.001$, $d = 2.860$) and Relative maximal loads (CRIMP *vs* SLOPER: $p < 0.001$, $d = 0.3446$; SLOPE *vs* SLOPER: $p < 0.001$, $d = 0.3545$) than the other grip positions.

When global muscle activity was compared, ANOVA results revealed a significant grip effect on $RMS_{Global}$, showing that greater global sEMG activity occurred during the SLOPER

**Table 2** Descriptive statistics for load, muscle activity, and neuromuscular efficiency variables.

| Variables | CRIMP<br>Mean ± SD | SLOPE<br>Mean ± SD | SLOPER<br>Mean ± SD |
|---|---|---|---|
| Maximal load (kg) | 94.12 ± 12.4 | 93.6 ± 11.8 | 133.8 ± 16.0[*] |
| Relative maximal load (%) | 133.3 ± 14.3 | 132.6 ± 13.7 | 189.5 ± 18.1[*] |
| $RMS_{Global}$ (mV) | 1751.79 ± 413.51 | 1619.25 ± 471.2 | 1865.06 ± 431.63[*] |
| $RMS_{FDP}$ (mV) | 708.96 ± 172.04 | 651.05 ± 188.39 | 701.52 ± 156.13 |
| $RMS_{FDS}$ (mV) | 610.64 ± 236.73[**] | 520.87 ± 228.28 | 675.56 ± 231.44[*] |
| $RMS_{FCR}$ (mV) | 189.77 ± 70.04 | 179.43 ± 79 | 292.6 ± 130.81[*] |
| $RMS_{EDC}$ (mV) | 242.42 ± 111.08 | 267.91 ± 234.66 | 195.39 ± 123.91[*] |
| $NME_{Global}$ (N/mV) | 0.56 ± 0.19 | 0.62 ± 0.2 | 0.75 ± 0.25[*] |
| $NME_{FDP}$ (N/mV) | 1.4 ± 0.46 | 1.58 ± 0.69 | 1.98 ± 0.61[*] |
| $NME_{FDS}$ (N/mV) | 2.1 ± 2.39[*] | 2.38 ± 2.11 | 2.39 ± 1.72 |
| $NME_{FCR}$ (N/mV) | 5.6 ± 2.33 | 6.25 ± 3.05 | 5.32 ± 2.42 |
| $NME_{EDC}$ (N/mV) | 4.55 ± 2.09 | 4.84 ± 2.29 | 8.62 ± 3.84[*] |

Notes.

Abreviations and symbols: RMS, Root Mean Square; FDP, flexor digitorum profundus; FDS, flexor digitorum superficialis; FCR, flexor capri radialis; EDC, extensor digitorum communis; NME, Neuromuscular efficiency.

[*]Significantly different from the other two grip positions.

[**]Significantly different from SLOPE grip only.

**Table 3** Comparison of load and muscle activity variables among grip positions using one-way repeated measures ANOVAs or Friedman's test.

| Variables | F | $\chi^2$ | df | p | $\eta_p^2$ | Kendall's W | Power | Post-hocs |
|---|---|---|---|---|---|---|---|---|
| Maximal load | 361.382 | – | 2,23 | <0.001 | 0.938 | – | 1.000 | SLOPER >CRIMP, SLOPE |
| Relative maximal load | 379.046 | – | 2,23 | <0.001 | 0.940 | – | 1.000 | SLOPER >CRIMP, SLOPE |
| $RMS_{Global}$ | 8.068 | – | 2,23 | 0.003 | 0.252 | – | 0.898 | SLOPER >CRIMP, SLOPE |
| $RMS_{FDP}$ | 3.161 | – | 2,23 | 0.074 | 0.116 | – | 0.464 | – |
| $RMS_{FDS}$ | 22.957 | – | 2,23 | <0.001 | 0.489 | – | 1.000 | SLOPER >CRIMP >SLOPE |
| $RMS_{FCR}$[a] | 24.238 | – | 2,23 | <0.001 | 0.502 | – | 1.000 | SLOPER >CRIMP, SLOPE |
| $RMS_{EDC}$[b] | 12.604 | – | 2,23 | <0.001 | 0.344 | – | 0.995 | CRIMP, SLOPE >SLOPER |
| $NME_{Global}$[a] | 28.452 | – | 2,23 | <0.001 | 0.542 | – | 1.000 | SLOPER >CRIMP, SLOPE |
| $NME_{FDP}$[a] | 34.137 | – | 2,23 | <0.001 | 0.587 | – | 1.000 | SLOPER >CRIMP, SLOPE |
| $NME_{FDS}$ | – | 18.960 | 2 | <0.001 | – | 0.379 | – | SLOPER, SLOPE >CRIMP |
| $NME_{FCR}$[a] | 1.820 | – | 2,23 | 0.173 | 0.070 | – | 0.361 | – |
| $NME_{EDC}$ | – | 31.920 | 2 | <0.001 | – | 0.638 | – | SLOPER >CRIMP, SLOPE |

Notes.

Abbreviations: FDP, flexor digitorum profundus; FDS, flexor digitorum superficialis; FCR, flexor capri radialis; EDC, extensor digitorum communis; NME, Neuromuscular Efficiency.

[a]Transformed *via* log10.

[b]Transformed *via* 1/x.

compared to the other grip positions (CRIMP *vs* SLOPER: $p = 0.044$, $d = 0.268$; SLOPE *vs* SLOPER: $p = 0.007$, $d = 0.544$) (Fig. 3A).

Individual muscle RMS comparisons also indicated a significant main effect of grip position on the $RMS_{FDS}$, $RMS_{FCR}$, and $RMS_{EDC}$ (Fig. 3B). Pairwise comparisons showed the greatest activity of the FDS (CRIMP *vs* SLOPER: $p = 0.005$, $d = 0.277$; SLOPE *vs* SLOPER: $p < 0.001$, $d = 0.673$) and FCR (CRIMP *vs* SLOPER: $p < 0.001$, $d = 1.049$;

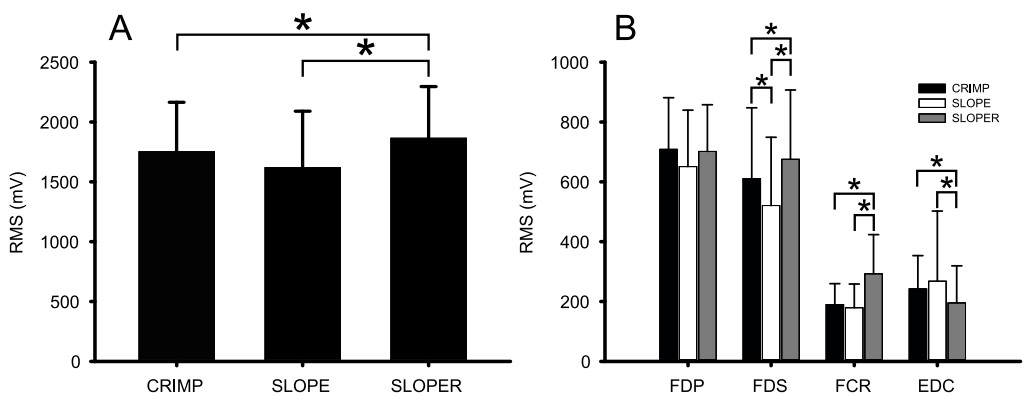

**Figure 3** Comparison of global sEMG activity (A) and individual muscle sEMG activity (B) among grip positions. FDP, flexor digitorum profundus; FDS, flexor digitorum superficialis; FCR, flexor carpi radialis; EDC, extensor digitorum communis. *Indicates significant differences ($p \leq 0.05$) between grip positions.

SLOPE *vs* SLOPER: $p < 0.001$, $d = 1.147$) in the SLOPER among grips. Conversely, the EDC showed less activity during the SLOPER compared to CRIMP ($p < 0.001$, $d = 0.668$) and SLOPE ($p = 0.005$, $d = 0.505$). In addition, the FDS also showed greater activity when using the CRIMP compared to the SLOPE grip ($p = 0.001$, $d = 0.386$).

Lastly, NME comparisons yielded significant grip effects for $NME_{Global}$, $NME_{FDP}$, $NME_{FDS}$, and $NME_{EDC}$. Pairwise comparisons showed that SLOPER presented the greatest values of $NME_{Global}$ (CRIMP *vs* SLOPER: $p < 0.001$, $d = 0.950$; SLOPE *vs* SLOPER: $p < 0.001$, $d = 0.629$), $NME_{FDP}$ (CRIMP *vs* SLOPER: $p < 0.001$, $d = 1.200$; SLOPE *vs* SLOPER: $p < 0.001$, $d = 0.777$), and $NME_{EDC}$ (CRIMP *vs* SLOPER: $p < 0.001$, $d = 1.315$; SLOPE *vs* SLOPER: $p < 0.001$, $d = 1.194$). Furthermore, CRIMP showed lower $NME_{FDS}$ values compared to the SLOPE ($p = 0.003$, $d = 0.125$) and SLOPER ($p < 0.001$, $d = 0.140$).

## DISCUSSION

The main findings of the present study suggest that the SLOPER could be a valuable grip condition towards finger strength training for climbers. SLOPER presented a greater global muscle activity, produced by a greater activity of the FDS and FCR muscles, in comparison to the CRIMP and the SLOPE grip. These results suggest that including the SLOPER in finger strength training programs should be advised, especially considering that its execution could result in lower soft tissue stress (*i.e.*, annular pulleys and skin) compared to the other grip positions. On the other hand, we also suggest that the CRIMP could potentially stimulate the FDS in a greater manner than the SLOPE, although its increased stress on the annular pulleys should be considered when prescribing this grip position.

### Global muscle activity

When global muscle activity was compared, we observed that the CRIMP and SLOPE grips presented similar muscle activity levels, whereas the SLOPER presented greater global activity compared to the other two grip positions. To our knowledge, only one study has assessed forearm muscle activity aiming to compare two or more grip positions

(*Watts et al., 2008*). *Watts et al. (2008)* compared six different grip positions, including the CRIMP and the SLOPE. Despite finding superior activity for the CRIMP, interpretation and comparison of their results is limited because they only measured sEMG from a single muscle on the anterior forearm and they did not equate intensity across grip positions. In addition, the study of *Watts et al. (2008)* did not assess forearm muscle activity in the SLOPER grip.

Our differences observed in global muscle activity could be caused by various factors. On one hand, it is well established that muscle activity assessed by sEMG amplitude is related to the magnitude of the muscle contraction and force or torque production (*De Luca, 1997*). Because we equated the relative intensity across conditions, the bigger usable surface of the SLOPER resulted in a superior mechanical advantage and allowed for higher loads, possibly favoring the production of higher sEMG in this grip position. The fact that SLOPER presented a higher NME indicates that this grip required lower muscle activity for each Kg of load used. Therefore, SLOPER can attain greater muscle activity, although the load necessary to reach this activity level is higher than for the other grip positions. On the other hand, differences in the gripping position itself could have caused changes in the contributory role of each of the joints involved, especially the interphalangeal and wrist joints (*Charissou et al., 2017*; *Caumes et al., 2019a*; *Beringer et al., 2020*). Therefore, changes in the activity of the muscle groups involved are presented below.

## Individual muscles: SLOPER *vs* CRIMP and SLOPE

The SLOPER grip also presented greater activity of the FDS and the FCR compared to the other two grips, while showing lower activity of the EDC. The FDP activity remained similar across the three grip positions. However, SLOPER also presented greater FDP, FDS, and EDC NME, indicating a relevant role of the used load on these muscles' activity.

Many climbing-related studies have focused on the role of the finger flexor muscles, as they are thought to be the primary agonist muscles during climbing and DHs (*Vigouroux & Quaine, 2006*; *Macleod et al., 2007*; *Schweizer & Hudek, 2011*; *Philippe et al., 2012*). FDP absolute activity was similar across grip positions, although SLOPER presented the highest FDP NME. Thus, comparable recruitment of the FDP could be achieved with any of the studied grip positions, but SLOPER may need higher absolute loads to achieve it. On the other hand, FDS showed increased activity in the SLOPER, indicating that this grip type can potentiate the use of an active flexion of the proximal interphalangeal joint (PIP) through the action of the FDS. Hence, high intensity training with the SLOPER could provide a greater stimulus for the FDS. However, this superior muscle activity could only be achieved at the expense of using a larger load than the one used with the other grip positons, as the SLOPER presented a greater FDS NME compared to the CRIMP. Although no previous research has compared the SLOPER grip with other grip positions in terms of differential muscle activity, we think that the differences found in the present study could be attributed to the necessity of maintaining friction with the curved sloper hold. To avoid slipping from SLOPER, the climber must maintain the center of pressure as close as possible to the flattest part of the hold; that is, parallel to the floor and close to the wall (*Fuss et al., 2013*). Because the sloper hold used in the present study had a depth of 60 mm, it allowed contact of the

middle phalanx with the hold positioning it almost parallel to the ground. We believe that these conditions could have facilitated an increase in the vertical force produced at the fingertip through active flexion of the PIP generated by the FDS action.

Similarly, the increased FCR activity observed in the SLOPER could also be attributed to the need to maintain sufficient friction with the curved hold. This need to optimize friction may require greater wrist stability, which is usually attained through co-contraction of the wrist flexors and extensors (*Caumes et al., 2019b*; *Caumes et al., 2019a*). In addition, the greater activity of the FDS while pressing on the hold with the fingers could generate a certain level of mechanical extension moment at the wrist, which might be counteracted by an increase in the activity of some of the wrist flexors, such as the FCR (*Charissou et al., 2017*). The fact that no differences in the NME were found between grips for the FCR may support this reasoning. Thus, differences in the muscle activity of the FCR could not be solely explained by a greater load during the SLOPER, but they also may be explained by the hand and fingers configuration and action during this grip type. The present FCR results underline the importance of using SLOPER grip positions to better stimulate the wrist flexors.

Regarding the EDC, its activity was lower in the SLOPER grip compared to the other two grip positions. It has been proposed that a co-contraction of the finger and wrist extensors may be used as a control strategy to increase wrist joint stability when performing gripping tasks (*Snijders et al., 1987*; *De Serres & Milner, 1991*). Thanks to this co-contraction, deviations from the desired position and an unintended wrist flexion moment generated by the finger flexors' action may be counteracted (*Snijders et al., 1987*; *De Serres & Milner, 1991*). However, different hand configurations during maximal gripping tasks may drastically affect the finger and wrist flexor-extensor co-contraction (*Charissou et al., 2017*). Specifically, *Charissou et al. (2017)* found that extensor activity was lower in a hand configuration consisting of finger-pressing on a surface similarly to our SLOPER grip, in comparison to power gripping. It was suggested that this difference in extensor activity was caused by a diminished need to counteract the unintended flexion moment generated by the finger flexors at the wrist. This hypothesis may support our results, where a greater activity of the FDS and FCR during the SLOPER would possibly facilitate maintenance of the center of pressure at the flattest part of the curved sloper hold while pressing on it. Thus, an opposition of the wrist flexion moment by the EDC might not be desirable, as observed by the lower EDC muscle activity and the greater efficiency we found in the SLOPER. Furthermore, previous climbing research using the SLOPE grip has proposed that the EDC may act alongside the FDS to stabilize the PIP joint, preventing FDP tension from driving this joint into hyperextension, a position also known as the swan neck position (*Schweizer & Hudek, 2011*). Although finger position during the SLOPE and the SLOPER might be similar, because the middle phalanx of the fingers was in contact with the hold during the SLOPER, it is unlikely that the PIP could hyperextend. Thus, lower stabilization of the PIP joint by the EDC might be required during the SLOPER.
### Individual muscles: CRIMP *vs* SLOPE

The CRIMP and SLOPE grips presented similar levels of activity in the FDP, FCR, and EDC, but the CRIMP showed greater FDS activity and lower NME. These results indicate that the CRIMP grip can better stimulate the FDS than the SLOPE, even when similar Maximal loads are used. Although no studies have examined the activity of the FDS between these two grip positions under intensity-equated conditions, others have explored the contribution of the FDS and FDP by analyzing force and tendon tensions (*Vigouroux et al., 2006*; *Schweizer & Hudek, 2011*). Our results are supported by a study performed with cadaver fingers where a more predominant role of the FDS was observed when using the CRIMP grip compared to the SLOPE grip on holds of a depth equal to the distal phalanx or greater. The results reported by *Schweizer & Hudek (2011)* support the greater FDS activity in the CRIMP grip observed in our study, which was performed on a 21 mm deep hold, approximately the length of the distal phalanx.

*Schweizer & Hudek (2011)* have hypothesized that when performing the SLOPE grip using a hold of about the same size of the distal phalanx (similar to our 21 mm hold), the FDS may play a stabilization rather than an agonistic role, preventing the PIP from assuming a hyperextended position (swan neck position), thus less FDS activity would be expected when using the SLOPE grip. Furthermore, *Schweizer & Hudek (2011)* observed greater efficiency (greater tendon-fingertip force transmission) of the FDS during the CRIMP compared to the SLOPE grip. If this hypothesis is correct, it is possible that climbers involuntarily prioritize use of the FDS when hanging using a CRIMP grip. Our results support these hypotheses when using the SLOPE grip on a 21 mm deep flat hold.

### Practical applications

Overall, our results may provide valuable information for coaches and practitioners as regards the design of climbing training programs. We suggest that the SLOPER has a greater recruitment potential of the agonist muscles (*i.e.,* FDS and FCR) than the CRIMP and the SLOPE. Therefore, inclusion of the SLOPER when training grip strength through DHs might be advised, especially for those seeking to stimulate the FCR. In addition, this grip position could diminish the stress of the soft tissue of the fingers (pulleys and skin) compared to the other two positions. However, greater loads might be needed for this improved recruitment compared to other grip positions, which may increase the stress in the shoulders and elbows and reduce the comfortability of the DHs, as it requires the addition of a lot of external load. Furthermore, to our knowledge no study has yet modeled the real impact of the SLOPER on the annular pulleys like it has been done with the other two grip positions. On the other hand, CRIMP grip also seemed to have greater training potential than the SLOPE grip, especially for the FDS. Moreover, because the CRIMP presented the lowest NME, greater FDS activity might be expected with equal or even lower loads compared to the other two grip positions.

### Study limitations and future directions

However, our study is not free of limitations. First of all, forearm is a difficult body region to measure sEMG because its muscles are rather small and are very close to each other. We

were extremely careful placing electrodes following indications from previous publications and checking the electrode placement by obtaining isolate muscle activity. However, it is possible that our sEMG signals, and hence our results, were affected by an inevitable level of crosstalk between muscles. Furthermore, we did not take anthropometric or kinematic measurements of the fingers and wrist joints and we were therefore unable to compute torques and further discuss on the extent of the effect of load on muscle activity or fingers soft tissue stress across conditions. Although our results could have several applications in climbing and training, they should be interpreted with caution because generalization of our results to different hold characteristics, such as shape or depth, or real climbing may not be possible. In addition, despite the fact that most of the participants performed with the same maximal loads in both sessions and we left a minimum of 3 days between sessions while asking the participants to refrain from training for 24 h before sessions, it is possible that some degree of fatigue or delayed onset muscle soreness affected the participants (especially the less trained ones). To completely avoid this confounding factor, we would recommend tracking the participants' activity to be sure that they really restricted their training activity for longer time periods. Lastly, our study mostly included intermediate and advanced level climbers with a wide range of anthropometric characteristics. Future studies should include anthropometrically homogenous groups of elite and higher elite level climbers to elucidate the possible effect of anthropometric characteristics and climbing expertise on forearm muscle activity.

## CONCLUSIONS

Our results have revealed that, when intensity was equated across grips, maximum loads, and global, FDS, and FCR muscle activity were higher in the SLOPER compared to the other two grip positions (CRIMP and SLOPE). However, the fact that the SLOPER also presented higher global, FDP, FDS and EDC NME, highlights the need to use higher loads when using this grip position. Moreover, the CRIMP presented greater FDS activity and lower NME compared to the SLOPE, and therefore could better stimulate this muscle group even when using similar loads.

## ACKNOWLEDGEMENTS

We thank Climbat Foixarda and its staff for helping with participant recruitment and implementation of the study. We also thank all the climbers who participated in the study and the undergraduate students who assisted during data collection.

### Funding

This study was supported by the Institut Nacional d'Educació Física de Catalunya (INEFC) of the Generalitat de Catalunya and the Grup de Recerca en Activitat Física, Alimentació i Salut (GRAFAiS, Generalitat de Catalunya 2021SGR/01190). The funders had no role

in study design, data collection and analysis, decision to publish, or preparation of the manuscript.

### Grant Disclosures

The following grant information was disclosed by the authors:

Institut Nacional d'Educació Física de Catalunya (INEFC) of the Generalitat de Catalunya and the Grup de Recerca en Activitat Física, Alimentació i Salut: GRAFAiS, Generalitat de Catalunya 2021SGR/01190.

### Competing Interests

The authors declare there are no competing interests.

### Author Contributions

- Blai Ferrer-Uris conceived and designed the experiments, performed the experiments, analyzed the data, prepared figures and/or tables, authored or reviewed drafts of the article, and approved the final draft.
- Denisse Arias performed the experiments, analyzed the data, authored or reviewed drafts of the article, and approved the final draft.
- Priscila Torrado performed the experiments, analyzed the data, authored or reviewed drafts of the article, and approved the final draft.
- Michel Marina performed the experiments, analyzed the data, authored or reviewed drafts of the article, and approved the final draft.
- Albert Busquets conceived and designed the experiments, performed the experiments, analyzed the data, prepared figures and/or tables, authored or reviewed drafts of the article, and approved the final draft.

### Human Ethics

The following information was supplied relating to ethical approvals (*i.e.*, approving body and any reference numbers):

The Ethics Committee for Clinical Research of the Catalan Sports Council granted Ethical approval to carry out the study (approval number: 25/2019/CEICGC).

### Data Availability

The data is available at Zenodo: Ferrer-Uris, Blai, Arias, Denisse, Torrado, Priscila, Marina, Michel, & Busquets, Albert. (2023). Effects of grip position on forearm muscle surface electromyography activity during maximal isometric finger dead-hangs in rock climbers [Data set]. Zenodo. https://doi.org/10.5281/zenodo.7597304.

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
