# Peer review of "Exploring forearm muscle coordination and training applications of various grip positions during maximal isometric finger dead-hangs in rock climbers"

_PeerJ, doi:10.7717/peerj.15464_

## Round 0.1 · original submission · Major Revisions

Reviewers have provided very detailed reviews of the manuscript that authors should consider to improve the quality of the manuscript.

·

Basic reporting

Review
Efects of grip position on forearm muscle surface electromyography activity during maximal isometric finger dead-hangs in rock climbers

Dear Authors, I really appreciate your work on this interesting topic. I find it useful for practicioners, physicians and scientists interested in the topic of rock climbers. Below I put my suggestions and recommendations on how to improve the quality of the paper.

Basic reporting
I find English professional and comprehensive, the flow is good. The background is well written. The references are up-to-date.
I recommend adjusting the text for better reading by the Editor and Reviewers.
Line 183 – add grip type for better understanding

Table 2 – please correct unit from (mv) to (mV) – check through whole manuscript.

I strongly recommend adding sections of Limitations of the study, Pracitcal application and Further studies directions.

Experimental design

Experimental design
Line 134 – the experimental protocol inform that the participants were not doing climbing training the day before, but does not inform about any other physical activities which can potentially affect the performance and the results. What is more, as is well known – the DOMS effect can last up to 5 or even more days especially in the unexperienced people (like in this study – training 2 times a week). I understand the limitations, but I recommend to: 1 use bigger pause from physical activity in further studies to exclude the doms effect 2 inform about this issue in the limitations of the study (unexperienced group and short rest period from earlier activities).

Figures
I have some observation to the figure 3 part A and B.
The lines used may suggest the reader that combined values differ significantly from another value – f.e – the reader may thing that in aspect of global RMS of sEMG combined crimp and slope differ from (single) sloper and not that (which is right) slope differ from sloper and crimp differ from sloper. In the part B the same issue is present. So I would recommend using separate lines to show statistical differences between particular results.

Validity of the findings

Validity of the findings
The variety of experimental group can be a limitation in this study – (big age differences - Age (years) 29.0 ± 7.3, big height differences - Height (cm) 170.7 ± 22.2 and very big bodymass differences - Body mass (kg) 75.1 ± 23.2 Climbing experience (years) 5.0 ± 3.3). In this study potentially a 22-yo women of below of 148cm, bw of below of 50kg and 2yr climbing experience training 1/week could be compared to a 36yr experienced man training 3-4 times a week with body height over 190cm and bm over 98kg. Other combinations are also possible (f.e a man with high BMI vs a woman with very low BMI) which may affect the transcutaneous resistance and sRMS values.
I recommend indicating the above in the limitations section and for further studies either choosing more consistent groups or selecting the consistent group and comparing the group results to the the extreme participants to show that the differences do or do not may have impact on the results.

Additional comments

I find the article good quality and professionally written, so I would recommend publication after minor corrections.

Reviewer 2 ·

Basic reporting

The manuscript reaches the standards of publication

Experimental design

I have some concerns with the methods that should be answered by modifying th etext and potentially doing additionnal data analysis:
1) I’m very sceptical in the possibility to distinguish clearly FDS from FDP EMG signal. Authors state that they validated the electrode placements when clear and isolated activity of each muscle was obtained. Given that FDS and FDP are synergist, obtaining a mechanical action of the one without the other is not possible. What kind of testing movements were used for that? Moreover, the forearm anatomy leads necessarily to crosstalk as FDP is positioned close to FDS. How was it managed? Which finger was targeted for the recording?
2) A photo is required to show the overall experimental design with a vision of the participants hanging with loads.
3) Filters used to clean the EMG signal seem atypical given the scientifical consensus: why 30hz as low cut off born instead of 10-20?
4) L175: p instead of alpha

Validity of the findings

Some comments on the results section:

-add some * in the table 1 to show the statistical significance. Otherwise readers should switch from table 2 to understand table 1
-F values of Anova should appear in the text

Additional comments

This study proposes to measure EMG of several hand muscles during various grip used in rock- climbing. The method and the results are interesting. The introduction and the discussion is well referenced and presents a good understanding/explanation of the phenomenon. Nevertheless, my major concern is the study positioning which, at this state is not well presented and should be modified. This will give to the manuscript the full potential of the conducted study and results. I hope my comments below will help:

Major Comments:
-Introduction
While the context and the bibliography associated to the impact of the different grip techniques is well presented, I have problem to understand the study justification/positioning.
Especially muscle activity is a measurement and is not a scientifical objective in itself. Muscle EMG is interesting since it gives information on the muscle state (level of involvements, level of fatigue etc. ). Assessing Muscle activity is thus not a valuable objective. Authors should expose more the real objective (e.g. understanding muscle coordination for training applications) which leads to the inherent use of EMG. By the way, the article title is also symptomatic of this positioning and should be modified.
Second, in spite of I well understand crimp/slope comparison with a similar hold, the justification of the Sloper is more difficult to understand: it is obvious that changing the shape of the hold will lead to changes in grip force and in muscle coordination since the surface of finger in contact with the hold are changed. This is well described by Amca 2012 (https://www.tandfonline.com/doi/abs/10.1080/02640414.2012.658845) and Amca 2014 (https://ojs.ub.uni-konstanz.de/cpa/article/view/6511). In this context, why choosing a Sloper? instead of a jug? instead of a larger ledge? instead of a finger lock? Etc?. There is an infinite number of hold shapes to explore. From my point of view, the authors should argue more WHY exploring Sloper is a valuable objective. Moreover from a pure scientific design point of view: comparing crimp/slope (a finger technique) while sloper (which is a shape and not a finger technique : the fingers are positioned in a slope posture type) is hardly arguable. The comparison between the 3 thus asks the logics of the brought knowledge.

-Discussion
I’m uncomfortable with the first paragraph of the discussion i.e.: the presented main findings. it is not a surprise that SLOPER generates more finger force than the other grip (bigger hold with more finger surfaces in contact) and which thus (as very well explained by the authors) allowed the muscles to generate more forces and thus more activities. In itself this obvious result should not be the main findings from my point of view. Otherwise, authors should conduct another study with a bigger hold (a jug as an exemple) to found similar conclusions and redo this demarche for each size and each shape of holds. I’m pretty sure that this study brings some knew and useful knowledge but the positioning of the study, with the objective of comparing the activity with different size hold/techniques is not the good way to present it. My suggestion is to focus the article on the benefit to train (or perform DH) on bigger hold (such as a sloper) to avoid training on small hold which could be source of injuries /source of skin pain. If authors can demonstrate that the muscle involvement (via EMG) is similar or superior, it gives argue for trainers and climbers to train on slopers rather than on a small rung. In itself this objective has a good meaning for application and justify why it is important to compare crimp/slope on small hold with a Sloper.

The rest of the discussion is very interesting and propose a very deep and accurate interpretation of the results. Congrats

Minor Comments:
-Introduction:
Add some references in the beginning of the introduction to justify the first sentence.

Discussion
Paragraph L302: this interpretation should be limited to the 21mm rung. Different sharing between FDS and FDP could appear with smaller/larger holds. This may explain the different conclusions between studies

·

Basic reporting

• Line 17 - Deadhangs are described as targeting the hand finger flexors. I do not believe hand is necessary in this description because the fingers imply the hand is involved. Therefore, for the reason of conciseness, I would recommend dropping hand from this.

• Lines 28-30: Provides a relevant summary of the results section, highlighting the most important findings from the researcher. However, this summary does not tell us whether the results are statistically significant or not. The author should consider including the statistics in this summary.

• Line 38-40: The author mentions that several studies have suggested that the ability to apply forces with the fingers is critical for rock climbing performance, whilst this is true they have cited one paper which is very general and compares climbers to non-climbers. For the claim they make it might be more useful to cite reviews by either Saul et al (2019) or Stein et al (2022). Although brief in nature these reviews will support the claim made, better than the paper they have cited.
o Saul, D., Steinmetz, G., Lehmann, W. and Schilling, A. (2019). Determinants for success in climbing: A systematic review. Journal of exercise science and fitness, 17(3), pp: 91-100.
o Stien, N., Saeterbakken, A. and Andersen, V. (2022). Tests and Procedures for Measuring Endurance, Strength, and Power in Climbing-A Mini-Review. Frontiers in Sports and Active Living. 4:847447. doi: 10.3389/fspor.2022.847447.

• Lines 43-46: Describes the common definitions for the slope and crimp grip. Yet they do not cite the literature where these definitions originate from. The author should consider reading and citing the paper by Schweizer (2001)
o Schweizer, A. (2001).Biomechanical properties of the crimp grip position in rock climbers. Journal of Biomechanics. 34, pp:217-223.

• Line 78: Explanations of what FCR and EDC stand for appear later in the paper after the abbreviations are originally used. The author should include the full names of the abbreviations in this line, like they do for the FDP and FDS earlier in the section.

• Line 380-381: Not in alphabetical order for references. Needs moving to line 374-375.

• Line 414-416: Not in alphabetical order. Needs moving.

• Line 426-428: Repeated references (same as the one prior). Does this need to be a different reference from Vigouroux et al?

Experimental design

• The description for the experimental protocol is long and hard to understand. For example, the maximal finger deadhang test is described in this section. For conciseness and ease of reading, the author should consider moving the description of the deadhang protocol to the data collection section. For an example of a clear and detailed methodological section, in climbing research, please read the paper by Giles et al (2020).
o Giles D, Hartley C, Maslen H, Hadley J, Taylor N, Torr O, Chidley J, Randall T, Fryer S. An all-out test to determine finger flexor critical force in rock climbers. International Journal of Sports Physiology and Performance. 2020;16(7):942-949.

• Lines 83-84: How was this sample size determined as being sufficient for power? Usually, A priori G∗Power 3.1 analysis is recommended to be used to calculate sample size requirements. Power analysis indicates that a sample size of at least 28 participants, resulting in a power of 0.8 (α = 0.05, f = 0.25). As neither the Vigouroux et al (2006) or Schweizer & Hudek (2011) papers provided effect sizes. This is based on the sample having sufficient power to find a medium effect size for an Anova (repeated measures; within factors).

• Line 83: Sample includes 22 males and 3 females. This is very imbalanced, and how were the sex differences controlled or considered in the data analysis? - Female subjects’ physiology should be seen as unique to male subjects, as such the research should avoid grouping female participants with male participants and assuming they are the same (Mujka and Taipale, 2019).
o Mujika, I. and Taipale, R. (2019). Sport Science on Women, Women in Sport Science. International Journal of Sports Physiology and Performance. 14, pp:1013-1014. doi.org/10.1123/ijspp.2019-0514.

• Line 99 and 251: The depth for the sloper hold is described as 60mm deep, however in figure 1E it is described as 600mm deep. Which one is it?

• Line 111-119: The maximal load DH protocol is described here. Whilst an adequate level of description is provided, it is difficult to understand the validity of this test because no literature is cited. Either a reference or an explanation on why this protocol was selected should be included.

• Whilst it is good that an experienced researcher controlled the form, it does not make this research repeatable. Thus, lacking reliability. Can you provide an explanation of the form used, whilst complete the deadhang? For an example of this please see either Torr et al (2020) or Giles et al (2020) papers (titled Rung and Test position or Hand and Body Positioning, respectively).
o Torr, O., Randall, T., Knowles, R., Giles, D., and Atkins, S. (2020). Reliability and validity of a method for the assessment of sport rock climbers' isometric finger strength. Journal of Strength and Conditioning Research.
o Giles D, Hartley C, Maslen H, Hadley J, Taylor N, Torr O, Chidley J, Randall T, Fryer S. An all-out test to determine finger flexor critical force in rock climbers. International Journal of Sports Physiology and Performance. 2020;16(7):942-949.

• Line 136-138: The description for the maximal DH protocol may be best placed here. Then follow on with how the score was recorded.

Validity of the findings

• Whilst the results are relevant and interesting to read, it is quite difficult to follow the narrative of the results section. The author should consider breaking the paragraph into smaller bitesize sections i.e., Global muscle activity and Individual muscles. Like in the discussion below. This will make it easier to navigate through the results section and understand what findings are trying to be communicated.

• For the discussion it is good practice to start with a general overview on whether the findings did or did not agree with the aims of your study. Once this has been established, then you can move into the specifics of the paper e.g., global muscle activity, individual muscles: crimp vs slope vs sloper, individual muscles: crimps vs sloper.

• Line 211-212: Says "Despite they found a superior activity for the CRIMP,". This does not flow nicely instead try something like the following; "Despite finding superior activity for the CRIMP".

• Line 214: change “no” to “not”.

• Line 224: Spelling mistake. Change to “positions”.

• Line 324: For the reason of conciseness drop “nevertheless”. It makes it too wordy when the point still stands without it.

Additional comments

The authors have chosen an original area of research to study and the findings of this research would be useful for both researchers and coaches. That said, the execution of the methods and results sections of this paper do not do the research the justice it deserves. I would strongly encourage the researchers to look at how other authors present their methods and results sections. They should consider breaking both the methods and results down into smaller bitesize sections that are easy to follow and digest the information. In the previous sections I have recommended papers which do this effectively. The authors should consider reading these papers to better understand how to structure both their own methods and results sections.

Once the edits have been made to the areas where the paper has failed to meet the standards of Peerj, I believe the paper will develop the research in climbing and have positively impact the area.

---

## Round 0.2 · accepted · Accept

The three reviewers are satisfied with the revised version of the manuscript. Congratulations on meeting the high standard publications of PeerJ.

·

Basic reporting

I appreciate the work of the Editor, other Reviewers and naturally the Authors to improve this manuscript. All my suggestions met the aproval of the authors and brought corrections in the manus. With great respect to other Reviewers I would recommend accepting this work for publication if the remarks pointed out by other reviewers are made by the authors in apropriate way.

Thank You all for cooperation and wish You good luck with Your further research.
Jarosław Muracki

Experimental design

all my suggestions were corrected

Validity of the findings

all my suggestions were corrected

Additional comments

all my suggestions were corrected

Reviewer 2 ·

Basic reporting

The authors did a great work by taking into account the reviewer’s comment. I’m very satisfied and I found this article suitable for publication and valuable for trainers and climbers. Congrats

Experimental design

The added Figure is valuable

Validity of the findings

Presentation of objectives and findings are more suitable

Additional comments

Minor comments :

-add a space between “positions” and “during” in the title
-P3L71: “When training finger strength, CRIMP or SLOPE grip positions are often used while manipulating the DH intensity by either varying the added load to the body mass or changing the deepness of the hold.” Could be interesting to note that load can also be subtracted to the body mass. As illustration Devise et al., 2022 (Devise, M., Lechaptois, C., Berton, E., Vigouroux, L. 2022. Effect of different hangboard training intensities on finger grip strength, Stamina, and Endurance. Frontiers in Sports and active living. 4, 862782) explored the effects of different force intensity trainings.
-p8 L257 : double space after “2016):”

·

Basic reporting

I am happy with the edits made by the authors.

The only thing I have noticed is that there is a small error in the title of the paper. On line 2 there is no space between positions and during. That said, it might just be how the pdf has loaded and I wanted to point it out just in case it was not.

Experimental design

I am happy with the edits made by the authors. No comments.

Validity of the findings

I am happy with the edits made by the authors. No comments.

Additional comments

I congratulate the authors on a detailed and thorough update to the manuscript. It is inspiring to see their positivity to take on all feedback, whilst having the confidence to challenge statements they felt were correct. I am satisfied with the responses to the concerns I have raised. I have no further comments.

Thank you for allowing me to be apart of this process.